# Teacher Burnout in Saudi Arabia: The Catastrophic Role of Parental Disengagement

**DOI:** 10.3390/bs13050367

**Published:** 2023-04-28

**Authors:** Georgios Sideridis, Mohammed H. Alghamdi

**Affiliations:** 1Boston Children’s Hospital, Harvard Medical School, Boston, MA 02215, USA; 2Department of Primary Education, National and Kapodistrian University of Athens, 15784 Athens, Greece; 3Department of Self Development Skills, Common First Year Deanship, King Saud University, P.O. Box 2454, Riyadh 11451, Saudi Arabia; mhalghamdi@ksu.edu.sa

**Keywords:** teacher burnout, cusp catastrophe, parent disengagement, teacher satisfaction, academic achievement

## Abstract

The present study predicts teacher burnout from previous experiences, efficacious beliefs, student achievement, and parental engagement. Data came from the Trends in International Mathematics and Science study (TIMSS 2019) and utilized a random sample of *n* = 2000 from the Kingdom of Saudi Arabia. It was hypothesized that parental engagement and involvement in school may represent a salient factor in understanding teacher burnout in that if parental disengagement is high, necessary supports and resources available to the teacher are withdrawn. This thesis was tested using the cusp catastrophe with linear negative predictors of teacher burnout being teacher satisfaction, years of experience, teacher efficacy, and student achievement. The role of parental disengagement was verified with critically low levels of parental engagement being associated with abrupt and unpredictable levels of teacher burnout. It is concluded that parental engagement and involvement in schools may provide critical supports that are necessary for teachers to successfully manage their workload.

## 1. Teacher Burnout: Consequences and Predictors

According to the World Health Organization’s (WHO) revised definition and the ICD-11 classification criteria, burnout is defined by three factors: increased pessimism about one’s job, decreased professional efficacy, and energy depletion and tiredness. The syndrome is described in more depth in the most recent definition, but what is more concerning is the increased prevalence of the disorder among teachers as compared to other professions, which has reached pandemic levels. Burnout rates have been reported to be 36% for K-12 teachers in Saudi Arabia, reflecting an alarmingly high rate [1].

According to some studies, burnout is the result of years of teaching combined with an incapacity to handle the demands of the job and excessive work demands [2]. There are numerous issues connected to teacher burnout. On a personal level, burnout is linked to emotional exhaustion [3], increased irritability [4], low motivation and autonomy [5,6], and poor well-being [7,8]. Teacher turnover [9,10], teacher absenteeism [11], the quality of instruction [12], low satisfaction [13], and low levels of English proficiency have all been linked to teacher burnout in the professional world. Students’ lack of motivation and autonomy, apathy, high or low receipt of disciplinary action, adversarial experiences with teachers, disruptive behaviors, increased depression, and low academic achievement are all student-related consequences of teacher burnout [14,15,16] and others [17,18].

Although prolonged exposure to stress may cause burnout, years of teaching have been suggested as a potential predictor of teacher burnout. Yet, empirical data have shown a more nuanced picture. For instance, in some studies, younger instructors with fewer years of experience had higher levels of burnout than older, more experienced teachers [19,20]. Ref. [21] emphasized that early in a teacher’s career, high levels of burnout are seen since new teachers are already worn out by extensive training, job outside of training, and family responsibilities. As a result of their lack of preparation for the job, newly appointed teachers may experience anxiety and burnout, according to [22]. For instance, disruptive student behavior has been proven to decrease newly appointed teachers’ engagement and commitment [23]. However, no variations in burnout were found between teachers’ experiences in other research. For instance, [24] found no difference in the level of burnout between new (less than 10 years of experience) and experienced instructors. Several studies found a curvilinear association between years of experience and teacher burnout, with a favorable second order term for years of experience [7]. Last but not least, interestingly, there are reported weak but negative relationships between teacher experience and burnout, suggesting that more experienced teachers are better at managing the demands of their jobs and have lower levels of burnout [25]. Burnout has also been linked negatively to job satisfaction, especially when school resources are scarce [26]. The importance of job satisfaction is found in its close associations with increased student accomplishment [5], fewer levels of anxiety and depression [27], and high levels of teacher efficacy and motivation. For instance, [28] found a link between student math achievement and teacher burnout (expressed as emotional tiredness). The beliefs of effectiveness among instructors may serve as a significant protective factor against burnout [29]. The ability of a teacher to achieve the appropriate levels of student engagement and learning, even with challenging or unmotivated students, has been defined as teacher effectiveness [30]. Efficacy, according to [31], also refers to a teacher’s capacity to manage interpersonal interactions with students and to carry out organizational duties (p. 684). The positive effects of teacher efficacy on classroom management [32], innovative instruction [33], effective instruction [32], increased persistence when working with struggling students [34], and students’ academic achievement were all supported by empirical evidence [35,36]. Worrying findings have also been revealed on the possibility that teachers may be more reactive and punitive toward children when poor efficacy is present [37].

Parental involvement at school. The academic life of students is significantly impacted by parents’ involvement in the school [38,39,40]. Parental participation typically manifests as parents having high expectations for their children’s academic performance and a dedication to supporting educational endeavors through collaborating with educators [41]. More specifically, parental resources and assistance in the form of time, money, and effort invested in helping students with their academic goals constitute parental engagement and involvement [42,43]. According to empirical research, parental involvement boosts students’ academic performance and motivation [44], teacher–student bonding [45], school readiness [45], attendance at school [45], graduation rates [46], and even their mental health and emotional development [47,48,49].

The mixed results were attributed by [47] to the nature of parent–teacher relationships, highlighting the need to view parent–teacher relationships as more complex than just the number of interactions because parents and teachers may have different perspectives on important aspects of their relationship [50,51]. In the parent–teacher relationship, for instance, [52] found a significant association between ethnicity and income. Others have linked parental marital status [53] or gender [54] to the strength of the parent–teacher relationship, with these characteristics affecting parents’ levels of confidence [55]. Last but not least, variations in the empirical literature may be the result of various methodological stances, research designs, and measuring tools [56,57]. Given the conflicting results of the pertinent literature on the function of parental involvement, the present study is an attempt to use a novel analytical approach to improve our knowledge of the nature of the relationship between parental engagement and teacher burnout.

## 2. Why a Nonlinear Framework?

The TIMSS currently uses solely quantitative methods to quantify parental participation. Yet, while taking into account the measurement of parent engagement in amount, a number of quality-related operations could go unnoticed. For instance, if parents are extremely interested and try to interfere, challenge, or exert control over the educational environment, high levels of parental engagement may be unsuitable. High levels of engagement with unfavorable feedback and confrontations with the teacher may be hazardous because they exacerbate tension and cause interruptions, which harms the learning environment and puts teachers under a lot of stress [58,59,60]. A high level of parental involvement, on the other hand, that encourages support for teachers and appreciates their knowledge and experience, where both sides collaborate to attain shared educational goals, would probably be adaptable. Parents may be seen as a valuable resource by teachers, and this collaboration may result in the development of a feeling of community and a positive school culture. The seeming paradox from a measurement perspective is that different teacher outcomes, whether positive or poor, may be associated with the same quantitative score in parental engagement. The requirement to precisely measure behavioral changes that are currently not possible to capture via the linear model is justified by the possibility that the same degree of predictor parental engagement may be associated with disparate and polar opposite results on teacher burnout. Low scores for parental engagement, which we shall refer to as disengagement, may potentially have implications for non-linearity. According to [61,62], it is possible that lack of parental involvement and engagement beyond a critically low level may significantly reduce the resources, supports, and means provided by parents, increasing the load, burden, and stress on the part of the teachers. As a result, increasing parent engagement past a certain point when it is extremely low may result in rapid and dramatic changes in the prevalence of teacher burnout. The cusp catastrophe model from the Nonlinear Dynamical Systems theory does a good job of capturing the latter [63].

## 3. Goals of the Present Study

The present study attempts to replicate the findings that have been predictive of teacher burnout such as years of experience, teacher efficacy, teacher satisfaction, and student achievement. Furthermore, the present study attempts to elaborate on the role of parental engagement and involvement with the school as a moderating factor that disrupts the linear relationship between the first set of factors and teacher burnout. It is hypothesized that when parental engagement with the school drops below some critical level, teacher burnout becomes unpredictable and chaotic. It is further proposed that non-linear modeling may better elucidate the relationships between complex phenomena such as teacher burnout.

## 4. Method

### 4.1. Participants and Procedures

Participants in the present study were drawn from the TIMSS 2019 4-grade survey in the Kingdom of Saudi Arabia. Analyses were produced using a random sample of 2000 participants as a means to control for the excessive levels of power that were associated with 7500 participants. However, results from both the full sample and random sample were literally identical; consequently, we present the results from the random 2000-participant group to avoid spurious statistical findings. Concerning the demographics, there were 957 females (49.7%) and 967 males (50.3%) (data on gender were missing from 76 participants (3.8%)). The mean teaching experience was 14.55 years (SD = 8.509). Age-wise, most teachers were 30–39 years (42%), followed by 40–49 years (38.1%) and 50–59 years (10.7%). The lowest frequencies were observed for the ages of 25–29 (8.2%) and less than 25 years (1%). The teachers majored in either mathematics (47%) or science (48.2%) with the remaining information being missing.

### 4.2. Measures

#### 4.2.1. Teacher Burnout

An 8-item scale of the TIMSS 2019 data under the label “About being a Teacher” was utilized as a measure of teacher burnout (see Appendix A). The items reflecting burnout were related to the number of students, the workload, the lack of time, and the teacher’s perceived pressure (see [64]). Items were scaled using a 4-point scaling system ranging from “Agree a lot” to “Disagree a lot”; thus, higher scores were indicative of low levels of teacher burnout.

#### 4.2.2. Teacher Efficacy and Parental Engagement and Involvement

Within the TIMSS 2019 school climate measurements, there is an 11-item scale that is completed by teachers and/or school principals. The scale has been termed “School emphasis on academic success” and involves items related to teachers’ efficacy, parental engagement, and students’ goals and attitudes toward school (see Appendix A). Items engage a 5-point Likert-type scaling system ranging from “very high” to “very low”, thus, high scores indicate lower levels of the respective latent traits. The developers of the instrument derived a total score assuming the unidimensionality of the three sources (teacher, parent, and student) resulting in a categorical system of low-medium-high emphasis on academic success. As Badri (2019) indicated, however, the teacher-form parent subscales are distinct, albeit positively correlated. Badri (2019), using CFA, pointed to the presence of a domain of teacher efficacy, and a parent domain that was termed parental engagement and involvement with the school. In the present study, we replicated the two domains that were produced by [65] using CFA methods. As shown in Appendix A, the parental engagement/involvement domain included items related to parents’ involvement, commitment, support, and expectations for students’ achievement. The analytical methodologies are described below.

#### 4.2.3. Teacher Satisfaction and Experience

Two items were engaged for the measurement of teacher satisfaction namely the ATDGTJS “Teacher’s job satisfaction” and teacher experience in years “ATBG01”. The measurement of job satisfaction followed the lead of [66] in proposing the measurement of overall satisfaction, rather than of specific job aspects.

#### 4.2.4. Student Achievement

The first plausible value in mathematics was utilized as an asymmetry variable. The correlation between the first and fifth plausible value estimates in math is over 0.99, so any selection will most likely lead to identical results.

### 4.3. Data Analyses

#### 4.3.1. Construct Composition: Confirmatory Factor Analysis and Omega Reliability

The constructs of teacher burnout, teacher efficacy on academic success, and parental engagement were constructed using the factor model in the form of factor scores if model fit was adequate and estimates of internal consistency reliability were acceptable. Consequently, unidimensional structures were tested using the 4-item teacher efficacy scale, the 4-item parental engagement scale, and the 8-item teacher burnout scale (see Appendix A). All models were run in Mplus 8.9, using the weighted least squares mean, and variance adjusted estimator (WLSMV) which is appropriate for Likert-type data. Evaluative criteria for these measurement models involved significant factor loadings for all items and descriptive fit indices such as the CFI and TLI above 0.900.

For the estimation of internal consistency reliability, we employed the Omega coefficient which is also most appropriate for non-tau equivalent measurements and reflects the ratio between true and total variance. Thus, relaxing the assumption that each item has an identical contribution to the measurement of the latent factor would most likely result in more accurate estimation of reliability, especially when this assumption is violated. In that sense, omega is superior to the popular Cronbach’s alpha indicator. It is estimated as follows:(1)$$\mathrm{Omega}=\frac{{\left(\sum^{\;}\lambda_{i}\right)}^{2}}{{\left(\sum^{\;}\lambda_{i}\right)}^{2}+\sum^{\;}\mathrm{Var}(\varepsilon_{i})}$$
with λ_i_ being the factor loadings of item i and ΣVar_(e)_ the respective error variances of item i.

#### 4.3.2. Cusp Catastrophe Model

Studies predicting teacher burnout have primarily relied on linear associations and have employed the linear model. It has long been observed, however, that under certain circumstances specific nonlinear processes are operative, which cannot be captured using linear models. One such model is the cusp catastrophe which describes qualitatively different states or behavioral modes. These distinct states, termed attractors, represent the behavioral space where the system moves in. Transitions between two behavioral attractors [67] are predicted by two control parameters, asymmetry (a), and bifurcation (b). The mathematical expression of the cusp engages a potential function *f*(*y*; a, b):*f*(*y*; a, b) = a*y* + 1/2b*y*^2^−1/4*y*^4^(2)

Equation (2) represents a dynamical system seeking optimization expressed by the first derivative (Equation (3)):*δ f(y)*/*δy* = −*y*^3^ + b*y* + a(3)

The response surface of the cusp model in Equation (3) (see Figure 1) depicts behavioral patterns as a function of two types of variables, termed a and b. At the back part of the upper surface, which signals low levels in the bifurcation variable, the behavioral change is smooth and predictable, best described using the linear model for the relationship between predictors and a single dependent variable (Pattern A to Linearity). However, when levels in the bifurcation variable increase beyond a critical level as in “Pattern B to Non-Linearity” in Figure 1, as depicted with “Point B”, the changes in the dependent variable will no longer be continuous. Thus, when levels in the bifurcation variable cross the critical point B (as in critically low levels of parental engagement), behavior will jump from the lower stable region to the upper stable region and the outcome (teacher burnout) will no longer behave in a linear manner as behavior will be oscillating abruptly between the two opposing behavioral modes (low–high). In the present context, where the shift is between the attractor of high burnout and the attractor of low burnout, it is considered that parental engagement, under certain conditions, may act as a bifurcation variable, in that lacking the support provided by parental engagement may act as a splitting control factor that leads teacher burnout to take on various, unpredictable levels.

The methodology involved in the estimation of the cusp catastrophe model used the cusp probability density function proposed by [68] using maximum likelihood estimation [69] and were executed using the package *cusp* in R. Evidence in favor of the cusp model, compared to the linear and logistic competing models would be manifested based on the following criteria as suggested by [69,70]: (a) significant difference using a nested difference chi-square statistic, favoring the cusp model, (b) lower AIC and BIC indices in favor of the cusp model, when contrasting non-nested models, (c) significance of both asymmetry and bifurcation variable terms in the cusp model, (d) bimodality of responses within the bifurcation area, (e) data within the bifurcation area being approximately 10% of the total number of data or comparison with proper competing models (in an effort to move away from arbitrary conventions, [70] suggested that a proper reference model is the logistic regression model whose sigmoid nature resembles the nonlinearity expected when the cusp catastrophe model is operative), and (f) presence of skewed distributions (alternating positive and negative skew) to the left and right of the bifurcation area in the cusp model.

## 5. Results

### 5.1. Prerequisite Psychometric Analyses of the Measured Constructs

All CFA models produced adequate model fit as judged upon two main criteria, (a) factor loadings significance and (b) descriptive fit indices values over 0.900. Specifically, the model fit for the teacher burnout scale was good with CFI = 0.948 and TLI = 0.927 and all measurement paths were significant (ranging from 0.344 to 0.787). Furthermore, omega reliability was 0.847, confirming its unidimensional structure. For teachers’ efficacy, again, all factor loadings were significant ranging in magnitude from 0.575 to 0.857. Model fit was excellent as shown using the CFI = 0.979 and the TLI = 0.936. Internal consistency reliability using omega was 0.728. Last, the parent engagement scale also possessed good psychometrics with factor loadings being significantly different from zero and ranging between 0.789 and 0.911, in standardized form. Descriptive fit indices were CFI = 0.998 and TLI = 0.994 with omega reliability being at 0.873.

### 5.2. Preliminary Results on the Relationship between Parental Engagemetn and Teacher Burnout

Initially, a series of linear and curvilinear models were fit to the data to examine the potential relevance of non-linear modeling towards predicting teacher burnout from the bifurcation variable, namely, parental involvement. Thus, linear, quadratic, and cubic models were run with the cubic model providing the best-fitted model as all terms, linear (*b* = −0.223, *p* < 0.001), quadratic (*b* = 0.140, *p* < 0.001), and cubic (*b* = −0.050, *p* < 0.001) were significant. The amount of variability explained by the cubic model was 12.7%. As shown in Figure 2, as parental disengagement increased, levels of teacher burnout increased (reflected as a negative slope because of the negative direction of both measures); for medium levels of parental engagement, the relationship appeared to be approximately null, and very low engagement was associated with a significant increase in teacher burnout. This visual presents the first evidence for a model that is very close to that of the cusp catastrophe. In fact, several analytical methods attempting to capture the cusp catastrophe model are based on polynomial regression [67].

### 5.3. Parental Engagement and Teacher Burnout: A Cusp Catastrophe Modeling Approach

These results are depicted in Table 1 with parameter estimates for intercepts and slopes for the four asymmetry variables namely, teacher efficacy, teacher satisfaction, years of experience, and student’s math achievement, the bifurcation variable parental disengagement, and teacher burnout as the outcome variable. Given that high scores on both parental engagement and teacher burnout signal the opposite, i.e., low levels of burnout and parental engagement, a negative sign in their bivariate relationship points to the expected direction that is, more parental engagement is associated with a lower likelihood for teacher burnout. As shown in the table, the bifurcation term was important suggesting that parental disengagement beyond some critical point was associated with chaotic levels of teacher burnout. The asymmetry variables were all significant and in the expected direction in that teacher efficacy (reflecting high expectations and success orientation) was associated with lower levels of teacher burnout and the same was true for teacher’s satisfaction, teacher’s years of experience, and students’ high levels of mathematics competency. A chi-square difference test favored the cusp model over its linear counterpart (χ^2^(2) = 212.800, *p* < 0.001). Furthermore, information criteria also favored the cusp model over the linear and logistic competing models as the estimates of AIC, AICc, and BIC criteria were consistently lower in the cusp model compared to the rest of the models (see Table 2). Figure 3 shows the density functions at various areas of the control space. As expected, skew and multimodality, and generally the absence of normality were observed across various areas of the response surface [70] with a small percentage of a bimodal distribution falling within the bifurcation area (grey). Last, Figure 4 shows that most observations lie on the upper and lower surfaces with some oscillating from the upper to the lower surface and within the bifurcation area. These are all expectations that the cusp catastrophe model holds.

## 6. Discussion

The purpose of the current research was to further investigate the impact of parental involvement and engagement in the classroom as a moderating factor that changes the linear link between adaptive processes such as student growth, teacher job satisfaction, efficacy, and burnout experiences. It is thought that when parental engagement with the school goes below a vital threshold, teacher burnout becomes unpredictable and chaotic, leading to a loss of control for the teachers. This hypothesis was put to the test using the cusp catastrophe model.

### 6.1. Cusp Model Interpretation

The application of the cusp catastrophe model led to the discovery of several significant findings. It first turned out to be statistically superior to both the linear model and the logistic model, which is closer to the cusp. Years of experience was one of the positive linear predictors of teacher burnout, indicating the existence of accumulated pressures over time. This conclusion differs from the majority of previous empirical studies in that there have been documented non-significant variations in burnout owing to experience [71]. The utilization of the cusp catastrophe model resulted in the identification of a number of findings that were of significant importance. In the beginning, it was discovered that it was statistically superior to both the linear model and the logistic model, which is located somewhat closer to the cusp. One of the positive linear predictors of teacher burnout was a teacher’s number of years of experience, which indicates the existence of pressures that have accumulated over time. This conclusion diverges from the findings of the vast majority of the earlier empirical studies in the sense that there have been documented variations in burnout that are not significant due to experience [71]. The functions of parental engagement as a bifurcation variable were supported more strongly by the cusp model. Fluctuations in teacher burnout are smooth and linear, characterized by high levels of professional engagement and low levels of the bifurcation variable. On the other hand, the system enters the nonlinear phase when the level of parental involvement reaches a critical threshold value (that is, when the level of parental involvement decreases as it is reversed coded). During this phase, a rapid switch can be seen between two behavioral attractors. As a consequence of this, the bifurcation point occurs at certain low levels of parental involvement, which we refer to as disengagement. Once the system passes this point, it enters the bifurcation set, which is the region in which discontinuous changes are observed (see Figure 4). Participants in this inaccessible area may find themselves being pushed in either the direction of high burnout or the direction of low burnout. There is a phenomenon known as hysteresis in which teachers who have the same level of parental involvement may or may not eventually exhibit signs of burnout. This is dependent on the personality of the teacher, the availability of resources, the presence of social support, and other factors.

### 6.2. Implications of the Findings for Educational Policy

The cusp model provided a greater amount of support for the functions of parental engagement as a bifurcation variable. There is a high level of parental involvement, there are only moderate levels of the bifurcation variable, and the variations in teacher burnout are both smooth and linear. On the other hand, once parental involvement reaches a critical threshold value, the system enters the nonlinear phase, which is characterized by a rapid transition between two behavioral attractors. This occurs when a child’s interaction with his or her parents reaches this value (i.e., parent participation drops as it is reversed coded). As a result, once some modest levels of parental participation are exceeded, the system enters the bifurcation set, which is the area where discontinuous changes are detected. This is what we mean when we use the term “disengagement” (see Figure 4). In this inaccessible location, particles may be pushed in either a high or low burnout direction, depending on which of the two is chosen. Teachers who encounter the same level of parental involvement may or may not eventually show signs of burnout. This is determined by a variety of factors, including the nature of the educator, the availability of resources, the existence of social support, and other similar considerations. This kind of behavior is referred to as hysteresis. In order for educational institutions to achieve their goal of increasing parental involvement, they will need to develop specific interventions to increase the level of trust between parents and teachers [55]. Within the context of these initiatives, a teacher may serve as a point of contact for activities that bring together parents and other educators. When directed to family advisory groups, school-wide programs that train parents on behavioral interventions, such as how to handle challenging behaviors at home, have the potential to boost participation as well as constructive interactions between parents and teachers, which will help build trust. After conducting the screening for problematic behaviors, [72] proposed “proactive outreach to parents” as a means of involving the parents in this process through the use of home-based interventions. Ref. [73] provided some suggestions for encouraging family participation by being sensitive to cultural differences and taking into account the unique circumstances of individual families. The widespread perception that instructors are taking on too much work represents yet another essential direction. Therefore, decision-makers in charge of policy may choose to concentrate their efforts on developing support mechanisms for teachers to assist them with tasks such as grading, assignments, and other types of preparation work. This position presents opportunities for schools, principals, and teachers to all be effective [74]. When it is absolutely necessary to do so, assistance can also include the provision of treatments related to mental health [74]. In conclusion, it is possible that it would be adaptive to implement teacher reward schemes in order to encourage engagement and prevent burnout. Examples of awards include ones given out for “teacher of the month” or “upon reaching a certain milestone” (such as student achievement concerning national standards, etc.).

### 6.3. Study Limitations and Directions for Future Research

The current study has significant limitations. First, as the study is correlational in nature, no conclusions about causality should be taken. Second, although having good psychometrics, some of the scales only had 4 to 8 items, which could have impacted content validity. Especially for the measurement of burnout, concerns have been raised on the presence of specific measurement errors due to the presence of specific personality traits such as neuroticism [75] and other personal dispositions. Third, opposed to the logic of linear modeling, the preference for a non-linear regime, such as the cusp model, argues that the moderating variable may be connected with two different and opposing outcomes. Fourthly, it would also be desirable to examine these relationships using additional non-linear multivariate models. Fifth, because the existing data did not include information on teacher perceptions given parents’ gender, the importance of dads in the parent–teacher interaction could not be understood.

To confirm that the current results are solid and applicable to the population of teachers in Saudi Arabia, it will be crucial to reproduce the findings in the future with new samples and older students. Additionally, the contemporary use of ecological systems theory can be broadened beyond the family and school to include larger systems including neighborhoods and communities based on racial, cultural, and religious considerations [76]. Furthermore, further research is necessary to understand predictor variables that are crucial from the perspective of educational policy, such as school and district quality indicators, as they may moderate the association between teacher burnout and student results [77].

## Figures and Tables

**Figure 1 behavsci-13-00367-f001:**
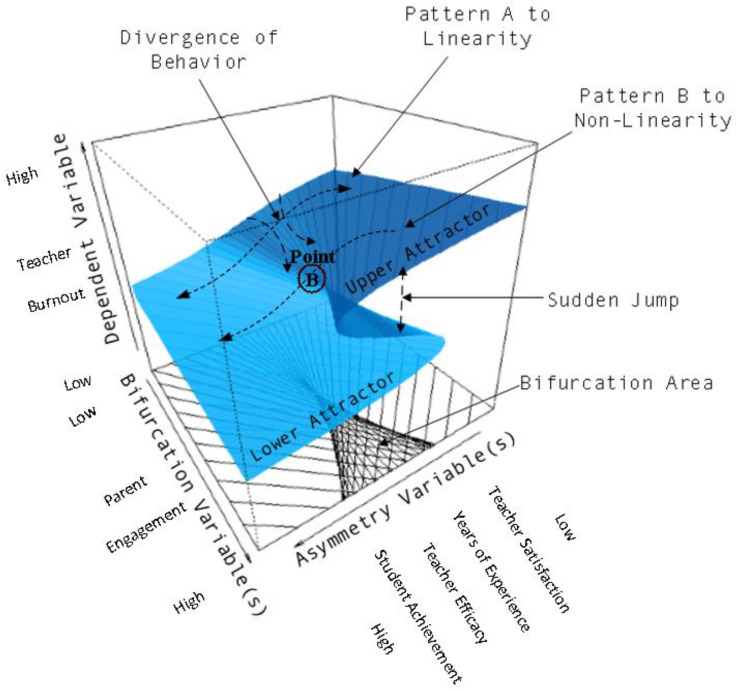
Description of the cusp model within the context of teacher burnout study. When asymmetry and bifurcation levels are low, the relationship between the focal variables is expected to be linear (Pattern A) and predictable. When levels in the bifurcation variable increase beyond a specific critical threshold, Pattern B is expected that is associated with non-linearity, unpredictability, and multimodality.

**Figure 2 behavsci-13-00367-f002:**
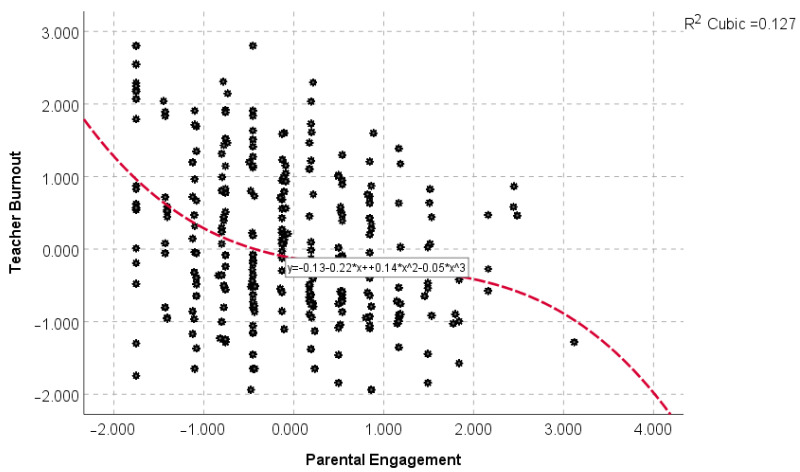
Evidence for a curvilinear relationship between parental engagement and teacher burnout.

**Figure 3 behavsci-13-00367-f003:**
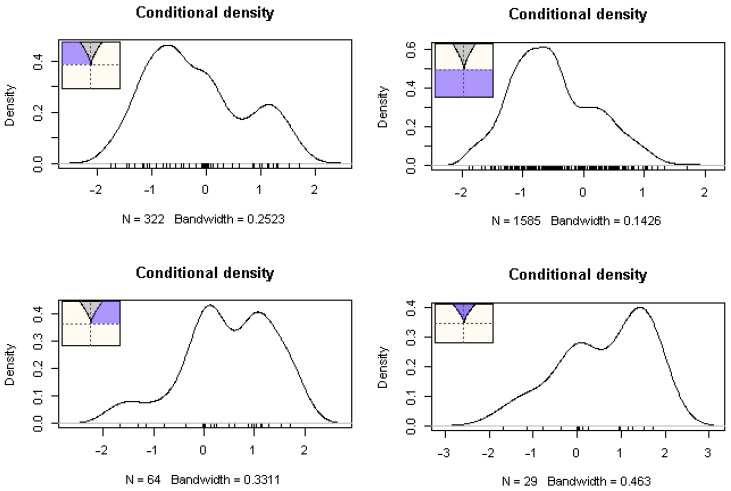
Conditional densities of observations at various locations on the response space surface. The presence of multimodality and skew are evident at various locations. Specifically, outside the bifurcation area, positively and negatively skewed distribution are expected (see upper left and lower left distributions). Within the bifurcation area non-normality and bimodality are also expected (bottom right distribution) with the expectation that up to 10% or less of the observations fall in this area.

**Figure 4 behavsci-13-00367-f004:**
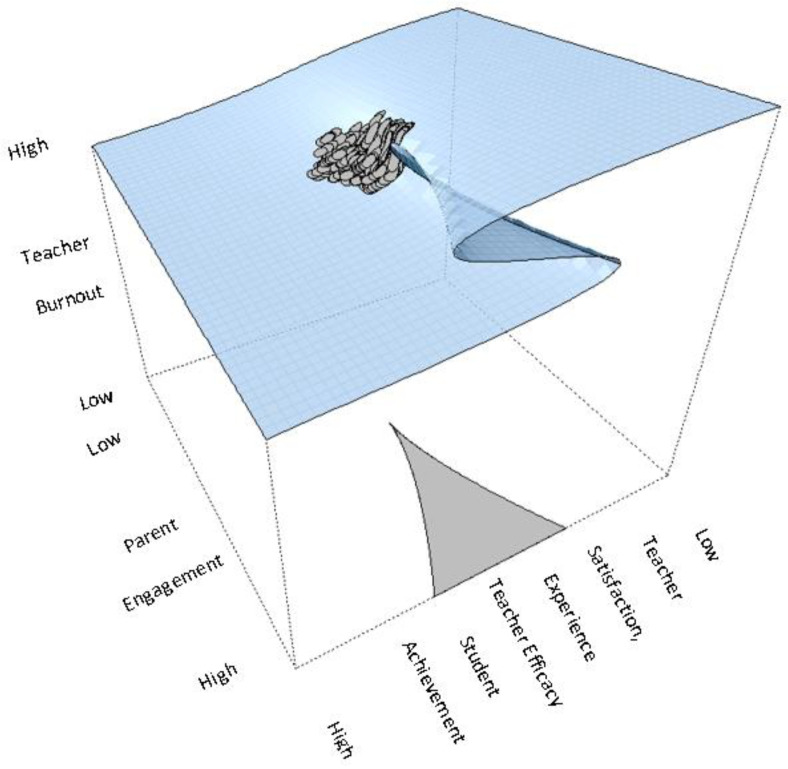
Cusp catastrophe model with observations moving from the stable upper attractor to the lower attractor, thus entering the bifurcation area of uncertainty and unpredictability. In other words, high levels in parental engagement (reversed coded in TIMSS) are expected to create uncertainty and unpredictability in the determination of teacher burnout.

**Table 1 behavsci-13-00367-t001:** Parameter estimates of the cusp model for the prediction of teacher burnout.

Terms in Cusp Model	B	C.I. B_95%_ Low	C.I. B_95%_ High	S.E.	*Z*-Value	*p*-Value
a (Intercept of asymmetry var.)	−0.123	−0.356	0.111	0.119	−1.031	0.303
a_1_ (Teacher’s Efficacy)	−0.332	−0.387	−0.276	0.028	−11.729	0.001 **
a_2_ (Teacher’s Satisfaction)	−0.557	−0.752	−0.361	0.100	−5.585	0.001 **
a_3_ (Teacher’s Years of Experience)	−0.243	−0.290	−0.196	0.024	−10.034	0.001 **
a_4_ (Student’s Math Achievement)	0.224	0.188	0.260	0.018	12.356	0.001 **
b (Intercept of bifurcation vars.)	−0.375	−0.454	−0.295	0.041	−9.255	0.001 **
b (Parental Involvement)	−0.418	−0.459	−0.376	0.021	−19.919	0.001 **
w (Intercept of outcome variable)	−0.418	−0.452	−0.385	0.017	−24.426	0.001 **
w (Teacher Burnout)	0.762	0.743	0.781	0.010	78.646	0.001 **

Note: ** *p* < 0.01. *p*-values were corrected using the Benjamini–Hochberg correction using a false discovery rate equal to the level of significance (5%). All values remained significant without the implemented corrective procedure.

**Table 2 behavsci-13-00367-t002:** Model comparison using information criteria.

Model Tested	Loglikelihood	Npar	AIC	AICc	BIC
1. Linear	−2706.65	7	5427.302	5427.358	5466.508
2. Logistic	−2629.42	8	5274.830	5274.902	5319.637
3. Cusp	−2600.25	9	5218.505	5218.596	5268.913

Note: Npar = Number of estimated parameters; AIC = Akaike criterion; AICc = Corrected AIC; BIC = Bayesian information criterion.

## Data Availability

Data are available from the official study of TIMSS 2019.

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
