# Peer review of "Teacher Burnout in Saudi Arabia: The Catastrophic Role of Parental Disengagement"

_behavsci, 2023, doi:10.3390/bs13050367_

Round 1

Reviewer 1 Report

Congratulations for your work. It shows an strong congruence among the different stages, the need for research is well explained, and the contribution of the Cusp Catastrophe Model results new to me.

Author Response

Thank you, we very much appreciated your positive response!

Reviewer 2 Report

The theoretical review is extensive and variables considered relevant to understand teacher’s burnout in SA. However, some key information is missing. For example, what is the TIMSS? An external reader cannot understand procedures without this background.

Secondly, how parental engagement was measured is not clear and the indicator of school achievement cannot be fully understood.

Finally, a sound argument regarding why burnout can be better measure with a cusp catastrophe model is not fully convincing.

Math complexity and sophistication cannot overcome methodological and logical flaws.

Author Response

Reviewer 2

The theoretical review is extensive and variables considered relevant to understand teacher’s burnout in SA. However, some key information is missing. For example, what is the TIMSS? An external reader cannot understand procedures without this background.

Answer

Thank you. We now present in the abstract prior to the abbreviation of TIMSS which is the “Trends in International Mathematics and Science Study.”

Secondly, how parental engagement was measured is not clear and the indicator of school achievement cannot be fully understood.

Answer

Thank you, all this information is presented in full detail in Appendix A where each construct and the respective items are shown, along with variable names and labels as they show up on the documentation files of the TIMSS study so that everybody can replicate our findings. Parental engagement is measured using the following four items that focus on parents’ involvement, commitment, expectations and support of students’ achievement.

Parental involvement in school activities

Parental commitment to ensure that students are ready to learn

Parental expectations for student achievement

Parental support for student achievement

We have now altered our text in the methodology to specify the content of this specific scale.

Last, for school achievement, the TIMSS psychometricians create standard scores for the measures of science and mathematics achievement. They usually provide several plausible values that can be used for future analyses. The plausible values are random draws from a posterior distribution of scores based on the observed response patterns on the assessment. We have now added in Appendix A that the correlation between plausible values was 0.99, so, using any plausible value should suffice to provide a valid proxy of students’ achievement in mathematics.

Finally, a sound argument regarding why burnout can be better measure with a cusp catastrophe model is not fully convincing.

Answer

Thank you, we have now revised our text on the section on “Why a nonlinear framework” to read as follows:

“The seeming paradox from a measurement perspective is that different teacher outcomes, whether positive or poor, may be associated with the same quantitative score in parental engagement. The requirement to precisely measure behavioral changes that are currently not possible to capture via the linear model is justified by the possibility that the same degree of predictor parental engagement may be associated with disparate and polar opposite results on teacher burnout. Low scores for parental engagement, which we shall refer to as disengagement, may potentially have implications for non-linearity.”

So, the main idea is that an outcome variable may be a function of some complex process that is not suitable to be captured using known analytical means such as linear, quadratic, cubic, and piecewise models. All these parametric models may fail to capture such a complex relationship that, theoretically speaking can be explained by bimodality or ,multimodality in outcomes.

Math complexity and sophistication cannot overcome methodological and logical flaws.

Answer

Thank you, this is true, and we agree with the reviewer 100%.

Reviewer 3 Report

Thank you for an interesting manuscript — overall, well done! There are however a few issues of varying import to address: 

* It's a small issue, but why are some words bolded? That is unusual for an academic article. It might be better to break these out as subsections to call attention to the ideas rather than bold them. 

* The title is nicely written — but needs to say that the study took place in Saudi Arabia. I suggest: "Teacher Burnout in Saudi Arabia: The Catastrophic Role of Parental Disengagement" 

* School systems, teaching training and parental involvement vary widely around the world. This manuscript needs a section that compares schools in the Kingdom and parental  involvement/disengagement to other major school systems in the world -- such as the United States, United Kingdom, Germany, Japan and elsewhere. Be systematic about the countries that you choose for comparison. Perhaps OECD countries. 

* To what extent are the study's findings applicable elsewhere? Or are they solely applicable in the Kingdom? If the latter, state so. If the former, justify how and why it would be applicable elsewhere. 

Author Response

Reviewer 3

Thank you for an interesting manuscript — overall, well done! There are however a few issues of varying import to address: 

* It's a small issue, but why are some words bolded? That is unusual for an academic article. It might be better to break these out as subsections to call attention to the ideas rather than bold them. 

Answer

Thank you, we have now eliminated all text that was bold. Initially we want to provide emphasis using bold but we don’t conform with the style of the journal and your suggestion and we thus, deleted any instance of bold text.

* The title is nicely written — but needs to say that the study took place in Saudi Arabia. I suggest: "Teacher Burnout in Saudi Arabia: The Catastrophic Role of Parental Disengagement" 

Answer

Thank you, we have now changed the title as suggested.

* School systems, teaching training and parental involvement vary widely around the world. This manuscript needs a section that compares schools in the Kingdom and parental  involvement/disengagement to other major school systems in the world -- such as the United States, United Kingdom, Germany, Japan and elsewhere. Be systematic about the countries that you choose for comparison. Perhaps OECD countries. 

Answer

Thank you, this is a very nice suggestion, but we believe it falls outside the scope of the present study. In the present study we only utilize data from the Kingdom of Saudi Arabia so any mention to comparison to other countries would be unjust because we did not employ any data from other countries. It is true that the present findings may not generalize to other countries, but this has never been our goal as the population of interest is only the Saudi Arabia Kingdom.

* To what extent are the study's findings applicable elsewhere? Or are they solely applicable in the Kingdom? If the latter, state so. If the former, justify how and why it would be applicable elsewhere. 

Answer

Thank you, indeed the findings only generalize to the Kingdom of Saudi Arabia given a national large representative sample. However, just because we did not test for our model generality in other countries, this does not mean that it cannot be the case with future studies. Thus, a future application could test the present methodology to other countries either within the Gulf area so that similarities can be identified or across all other countries involved in the TIMSS assessment. This would be an important study to verify whether the current methodology is applicable to other countries or is applicable and appropriate for the idiosyncrasies of the Saudi Arabia Kingdom. We are not currently able to tell, but we are hoping future research can inform this important suggestion.

Reviewer 4 Report

Thanks for the opportunity to review the article.

The article is very interesting and it adds to the existing knowledge in the field with some new highlights. 

However, I have a few small comments:

How did the authors treat repeated tests, and what is the level of significance with the necessary correction?

The authors stated: "The lowest frequencies were observed for the ages of 25029". Did they mean 25-29?

Results section: There is room to simplify the results section and explain the figures more clearly.

the next comment refers to the discussion. I think that the discussion is short and it could be developed in a more effective way that would highlight the results in a better way.

Author Response

Reviewer 4

Thanks for the opportunity to review the article. The article is very interesting and it adds to the existing knowledge in the field with some new highlights. However, I have a few small comments:

Answer

Thank you for your positive evaluation of our work.

How did the authors treat repeated tests, and what is the level of significance with the necessary correction?

Answer

Thank you, we have now added the Benjamini-Hochberg correction and we have added the following text below table 1.

“P-values were corrected using the Benjamini-Hochberg correction using a false discovery rate equal to the level of significance (5%). All values remained significant as without the implemented corrective procedure.”

The authors stated: "The lowest frequencies were observed for the ages of 25029". Did they mean 25-29?

Answer

Yes, thank you for your suggestion. We corrected it.

Results section: There is room to simplify the results section and explain the figures more clearly.

Answer

Thank you, we wish there would be a simple way to describe complex results. This is unfortunately not the easiest thing to do because the analytical methodology comes from physics and nonlinear analytical frameworks that most of social scientists have some difficulty in understanding fully. Nevertheless, we made honest attempts to explain the findings more fully.

the next comment refers to the discussion. I think that the discussion is short and it could be developed in a more effective way that would highlight the results in a better way.

Answer

Thank you, we expanded the discussion section with more content and also proofread our write-up so that it is more easily accessible to a wider audience.

Round 2

Reviewer 2 Report

Ammendements have clearly improved the manscript! congratulations.

Author Response

Thank you,

we very much appreciate your positive evaluation of our work!

Reviewer 3 Report

Thank you for a much-improved manuscript! However your insistence on not comparing your results to that of other countries makes the article less useful. If your intention indeed is to "only utilize data from the Kingdom of Saudi Arabia" then that needs to be made much more clear throughout the article, and particularly in the introduction and in the conclusion. For example, the article states that "Burnout rates have been reported to be 30 52% for K–12 teachers, higher by 22% over the average for all other occupations, which 31 was 30%" — is that in Saudi Arabia? If so, say so. And if not then it should not be cited unless you're willing to compare the burnout situation in Saudi Arabia to that of other countries. And if you are willing to do that, then please do so, as that is what will make this a useful article for people elsewhere in the world. 

Author Response

Thank you, we have now included only a reference to the Kingdom of Saudi Arabia with rates reflecting burnout in Saudi Arabia only. This is reflected using track changes.

Thank you for your suggestion to include more countries, but currently, the manuscript is already too large to expand it with more countries. We will certainly consider your suggestion as we develop more research works.

Again, thank you for your evaluation!

Reviewer 4 Report

Thank you for the revised version.. well done.. despite the fact that the results section still complex.. 

Author Response

(The authors gave the same response as above.)
